



# Significant seasonal changes in optical properties of brown carbon in the mid-latitude atmosphere

Heejun Han[1], Guebuem Kim[1], Kyung-Hoon Shin[2], Dong-Hun Lee[2]

[1]School of Earth and Environmental Sciences/RIO, Seoul National University, Seoul, 08826, Korea
[2]Department of Marine Sciences and Convergent Technology, Hanyang University, Ansan, 15588, Korea

*Correspondence to*: Guebuem Kim (gkim@snu.ac.kr)

**Abstract.** Atmospheric brown carbon (BrC) plays significant roles in the light absorption and photochemistry of the atmosphere. Although the occurrence and sources of BrC have been studied extensively, its removal processes and optical characteristics in the atmosphere have been poorly understood. In this study, we examined the seasonal changes in sources
and sinks of BrC and water-soluble organic carbon (WSOC) in the atmosphere of Seoul, Korea. Our results showed that the concentrations of BrC and WSOC decreased by approximately 80 % and 30 %, respectively, from the cold season (Oct–Jan) to the warm season (Jun–Sep). Excitation–emission matrix (EEM) spectra showed that the humic-like substance (HULIS) was the dominant fraction of BrC as the other components were not measurable. The air mass back trajectories of fire burning practices and the variations in K and V contents in the water-soluble aerosols during all seasons showed no
measureable decrease in input of biomass-burning sources in summer. However, there was a significant shift in photo-resistivity of light-absorbing organic aerosols in the summer, indicating significantly larger removals of ultraviolet (UV) degradable BrC. This was confirmed by laboratory UV radiation experiments on the optical property changes of BrC and WSOC in aerosol samples. Thus, our results suggest that the photo-degradation has dominant roles in controlling the quantity and quality of light-absorbing organic aerosols in the different seasons in the mid-latitude atmosphere.

## 1 Introduction

Organic aerosols play a significant role in atmospheric chemistry and global climate system (Kirillova et al., 2014a; Duarte and Duarte, 2013; Ghan and Schwartz, 2007). Most organic aerosols consist of a significant fraction of carbonaceous organic aerosols absorbing radiation (Laskin et al., 2015). These light-absorbing organic aerosols (or light-absorbing carbonaceous aerosols) including black carbon (BC) and brown carbon (BrC) have attracted increasing attention owing to their significant
roles in the radiative forcing of the global climate system by directly absorbing solar radiation (Kirillova et al., 2014a; Andreae and Gelencser, 2006; Graber and Rudich, 2006; Saleh et al., 2014; Ramanathan et al., 2007) and indirectly acting as cloud condensation nuclei (CCN) for cloud formation (Kirillova et al., 2014a; Andreae and Gelencser, 2006; Graber and Rudich, 2006; Kanakidou et al., 2005). BC is the most commonly known light-absorbing aerosols, which absorbs solar radiation over a wide spectral range, from ultraviolet (UV) to near-infrared radiation, while BrC highly absorbs radiation in





the range of UV to visible wavelengths (Feng et al., 2013; Laskin et al., 2015). Although the light-absorbing property of BrC is expected to be considerably weaker than that of the BC, its contribution would be significant due to its higher abundances over source regions (Hoffer et al., 2006; Gustafsson et al., 2009; Feng et al., 2013). Previous studies have estimated that the BrC contributes to approximately 19 % of the total atmospheric radiative forcing (Feng et al., 2013).

In general, BrC is known to originate from various sources such as biomass burning, incomplete combustion process, secondary formation, volatile organic compounds, and soil humics (Andreae and Gelencser, 2006; Graber and Rudich, 2006; Hoffer et al., 2006; Ramanathan et al., 2007; Kivácsy et al., 2008; Saleh et al., 2014; Laskin et al., 2015; Jiang et al., 2019). The origin of BrC is attributed predominantly to atmospheric humic-like substance (HULIS) (Graber and Rudich, 2006; Andreae and Gelencser, 2006; Laskin et al., 2015; Yan and Kim, 2017). In general, the atmospheric HULIS, contributes to
approximately 70 % of the water-soluble organic carbon (WSOC) (Laskin et al., 2015; Park and Son, 2017), while the WSOC contributes to 10–80 % of the total organic carbon contents in aerosols (Kirillova et al., 2014a; Kirillova et al., 2014b; Fu et al., 2015). However, current understanding of the optical properties, chemical compositions, and degradation processes of light-absorbing organic aerosols still remain uncertain.

In this study, we evaluated the seasonal changes in optical and chemical characteristics of organic aerosols in an urban region and changes in photo-resistivity of light-absorbing organic aerosols in the atmosphere. Recently, the method of excitation–emission matrix (EEM) characterization combined with a parallel factor analysis (PARAFAC) model has been employed for BrC studies (Kieber et al., 2006; Mladenov et al., 2011; Matos et al., 2015; Chen et al., 2016; Yan and Kim, 2017). This unique multi-analysis method can be used to identify individual chromophores of aerosol samples (i.e., HULIS
versus protein-like substance) and light-absorbing properties of each sample. In addition, the stable carbon isotope ratios of WSOC ($\delta^{13}C_{WSOC}$) and various chemical constituents were measured to identify the potential sources of organic aerosols in different seasons. Furthermore, we conducted laboratory experiments on the direct photochemical degradations of aerosol samples to verify the impacts of the UV radiation on the optical properties and carbon compositions of BrC and WSOC in the atmosphere.


## 2    Experimental methods
### 2.1  Study site and sample collection

Seoul is the capital city of South Korea and one of the largest metropolitan cities in the world. Korea has been highly affected by severe dust storms known as Asian dust or the Yellow dust originated from the Chinese and Mongolian deserts
during the spring (Mar–May) and often during the winter (Dec–Jan) (Lin et al., 2012). The increase in contents of fine aerosols associated with anthropogenic emissions is of great concern and a focus of major environmental studies in this region (Seinfeld et al., 2004; Park et al., 2007).



Aerosol samples ($N$=78) were collected using a high volume air sampler (HV-1000, SHIBATA) from March 2015 to January

2016 in Seoul, Korea (37.5° N, 127.0° E; 20 m above ground level) (Fig. 1). The samples were collected for 24 h at a

constant flow rate of 1000 L min$^{-1}$ through a pre-combusted (450 °C for 5 h) glass microfiber filter (GF/F, 8 × 10 inch, 2 μm

pore size, Whatman). A blank sample was collected by shortly exposing a blank filter at the study site and analyzed in the

same manner as that for the other samples. The total suspended particulates (TSP) of the aerosol samples were measured by

using the mass differences in desiccated filters between the pre- and post-sampling. The collected samples were covered with

an aluminium foil, placed in a polyethylene bag, and stored in the dark at -20 °C.

Meteorological parameters of the study site including the temperature and UV radiation rate were obtained from the ambient

air quality monitoring network named AirKorea in the Korea Environmental Corporation (KECO) and Korea Meteorological

Administration (KMA). Fire maps around the study site were obtained by using the moderate resolution imaging

spectroradiometer (MODIS) fire location data provided by NASA's fire information for resource management system

(FIRMS) (Fig. S1).

A ten-day air mass back trajectory was drawn by using the Hybrid Single-Particle Lagrangian Integrated Trajectory

(HYSPLIT) model to determine the source regions and transport pathways of air masses to the study site (Stein et al., 2015)

(Fig. 1). Due to a regional meteorology of the study site, which is dominated by the East Asian Monsoonal effect, most air

masses are transported from the arid and semi-arid regions in the Asian continent for most of the year (Yan and Kim, 2012;

Yan and Kim, 2017)

### 2.2 Aerosol extraction and chemical analyses

For the analyses of water-soluble organic aerosols, a small portion of filter paper was cut into small pieces and placed in a

pre-HCl-rinsed bottle. The organic components in the sample were extracted using Milli-Q water (18.2 MΩ cm) shaken at

125 rpm for 4 h (Wozniak et al., 2012). The extracts were filtered through a syringe filter (0.45 μm pore size Nucleopore,

Whatman) and analyzed to evaluate the WSOC and total dissolved nitrogen (TDN) by a high-temperature oxidation method

using a total organic carbon (TOC) analyzer (TOC-V$_{CPH}$, Shimadzu). The major ion species (NO$_X$, NH$_4^+$, SO$_4^{-2}$, Ca$^{2+}$, Cl$^-$,

Na$^+$, and K$^+$) were analyzed by using a high performance liquid chromatography (HPLC) (Waters 2695 system) system

equipped with a conductivity detector (Waters 432) (Yan and Kim, 2015). The non-crustal potassium (K) fraction was

calculated by using following equation: [nc – K] = [K] – ([K]/[Al])$_{crust}$×[Al]$_{aerosol}$ (Taylor and McLennan, 1995). The water-

soluble organic nitrogen (WSON) concentration was calculated by using the concentration difference between the TDN and

sum of inorganic nitrogen species (NO$_2^-$, NO$_3^-$, and NH$_4^+$). The sea-spray fraction was calculated by using Cl$^-$ and Na$^+$

concentrations assuming that all Cl$^-$ and Na$^+$ originated from seawater: sea-spray = Cl$^-$ + 1.4468 Na$^+$ (Maenhaut et al., 2008).

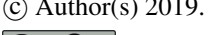



The subsamples for trace element analyses were dissolved in ultra-pure $HNO_3$ and analyzed by using a high-resolution inductively coupled plasma mass spectrometer (HR-ICP-MS, Thermo Element 2). The non-crustal vanadium (V) fraction was calculated by using the following equation: $[nc - V] = [V] - ([V]/[Al])_{crust} \times [Al]_{aerosol}$ (Taylor and McLennan, 1995; Yan and Kim, 2012). The molecular marker compound, levoglucosan, was measured by using a gas chromatography-mass

spectrometry (GC-MS) (7890N, Agilent) system coupled with a fused silica capillary column (HP-5MS, 25 m length, 0.25 mm i.d. and film thickness of 0.10 μm). Monthly representative filter samples were used for the analysis ($N=25$).

The HULIS fraction was separated by solid-phase extraction (SPE) by using DEAE column (GE Healthcare®, HiTrap™ DEAE FF, 0.7 cm ID × 2.5 cm) to validate the humic fraction obtained by using the EEM-PARAFAC results (Baduel et al.,

2009). The sample solutions were passed through the column at a constant flow rate of 1.0 mL min⁻¹. The neutral components and hydrophobic bases, and mono-, di-carboxylic acids, and inorganic anions were removed with Milli-Q water and 0.04 M NaOH, respectively. The polycharged compounds, HULIS, were eluted with 1M NaCl. The final fractions were analyzed for the HULIS quantification using a TOC analyzer.

The value of $\delta^{13}C_{WSOC}$ was measured by using an isotope ratio mass spectrometer (IRMS) (Isoprime, Elementar) combined with a TOC analyzer (Vario TOC cube analyzer, Elementar) (Panetta et al., 2008; Troyer et al., 2010; Kim et al., 2015; Yan and Kim, 2017). The isotopic composition $\delta^{13}C_{WSOC}$ was determined using the following equation (1):

$$\delta^{13}C = \left( \frac{(^{13}C/.^{12}C)_{Sample}}{(^{13}C/.^{12}C)_{Standard}} - 1 \right) \times 1000 \text{ ‰,} \tag{1}$$

where Vienna Pee Dee Belemnite (VPDB) was used as the isotope standard (Troyer et al., 2010; Fu et al., 2015; Kelly et al.,

2005). Analytical tests were made with IAEA-CH6 sucrose ($\delta^{13}C$ = -10.45 ± 0.03 ‰) and Suwannee River Fulvic Acid (SRFA) ($\delta^{13}C$ = -27.6 ± 0.12 ‰; International Humic Substances Society) to evaluate the recovery and the accuracy and of the measurements (Panetta et al., 2008; Troyer et al., 2010).

Fluorescence EEM and absorbance spectra of the aerosol samples were measured by using a spectrophotometer (Aqualog,

Horiba). The emission and excitation wavelength ranges were 240 to 700 nm and 250 to 500 nm, respectively, with scanning intervals of 1 nm. The PARAFAC model was performed using the Solo software in order to determine the fluorescent components in aerosols. The number of unique components was identified from the combined EEM data of aerosol samples. The results were validated by split-half analysis and analysis using random initialization (Bro, 1997; Stedmon and Bro, 2008; Zepp et al., 2004). The EEM results were normalized with Raman peak area of water and stated as Raman Unit (RU)

(Lawaetz and Stedmon, 2009). Raman and Rayleigh scattering signals, inner-filter effect, and blank subtraction were corrected using the Solo software.

The absorption coefficient was calculated using the following equation (2):



$$\alpha = 2.303\, A/l, \tag{2}$$

where $a$ is the absorption coefficient ($m^{-1}$); $A$ is the absorbance; and $l$ is the optical path length of the quartz cuvette (m). The spectral slope ($S_{\lambda o - \lambda}$) was calculated using following equation (3):

$$a_\lambda = a_{\lambda_{ref}} e^{-S(\lambda - \lambda_{ref})}, \tag{3}$$

where a is the absorption coefficient ($m^{-1}$); $\lambda$ is the wavelength; and $\lambda_{ref}$ is the reference wavelength (Twardowski et al., 2004; Helms et al., 2008). In this study, two distinct spectral ranges were selected in the shorter wavelength (275–295 nm) and the longer wavelength (350–400 nm).

### 2.3 Photochemical degradation experiments for light-absorbing organic aerosols

Laboratory experiments were conducted for two different durations using small portion of the aerosol filter samples inside of the incubator equipped with internal UV lamp (UV-B; 280–315 nm; 15 W). The filter samples for the experiment were duplicated or triplicated to rule out any experimental bias. The incubator was maintained under positive air pressure with a constant temperature at 20 °C. Aerosol samples were directly exposed to the stimulated UV radiation as its original particulate form, and each of the non-irradiated aerosol samples was evaluated for the initial conditions. For a long-term test, two representative aerosol filter samples were randomly selected from each of the higher and lower fluorescent groups, which had a similar range of WSOC concentration (summer = 16 μM; winter = 19 μM). After irradiation, filter samples were completely covered to prevent exposure to any light and directly extracted using the same procedure as described above. Each sample was analysed for optical and chemical properties after the irradiation.

### 3    Results

### 3.1   BrC compositions

The PARAFAC model identified two humic-like fluorescent components and one protein-like fluorescent component (Fig. S2). The spectral characteristics of component 1 (C1; Ex/Em = 305/416 nm) and component 3 (C3; Ex/Em = 365/484 nm) are known to be highly associated with the atmospheric HULIS (Chen et al., 2016; Yan and Kim, 2017). Component 2 (C2; Ex/Em = 290/340 nm) is associated with a protein-like (tryptophan-like) component, which originats mostly from biological production (Birdwell and Engel, 2010; Coble, 2007; Yan and Kim, 2017) (Table. S1). However, C2 seems to be largely influenced by the dominant fluorophore HULIS, as shown by the EEM spectra. A good correlation between C1 and C2 ($r^2$ = 0.9; p < 0.05) was obtained, although, in general, these two components have different sources and sinks in the atmosphere (Yan and Kim, 2017) (Fig. S3). Thus, we assumed that the HULIS (C1 and C3) was the dominant component of BrC in these samples, and thus C1 was used as a representative component of BrC in this study, as C1 and C3 exhibited a good correlation ($r^2$ = 0.8; p < 0.05) (Fig. S3).





The HULIS component obtained by the EEM-PARAFAC analysis agreed very well with the extracted HULIS concentration obtained by using the DEAE column ($r^2 = 0.81$; $p < 0.05$), which indicates that the C1 component represents the actual HULIS (Fig. S4). In addition, we compared the extraction efficiencies of HULIS (water-soluble BrC) to that of the MeOH-soluble BrC (Fig. S4). Although the efficiency of the water-soluble BrC was approximately 20 % lower than that of the

MeOH-soluble BrC, they exhibited a good correlation ($r^2 = 0.9$; $p < 0.05$). Therefore, we conclude that HULIS is a good representative component of BrC in this study.

### 3.2 Seasonal variations in WSOC and BrC

The WSOC concentrations exhibited seasonal variations in the range of 3 to 40 $\mu g\ m^{-3}$ (average = 16±7 $\mu g\ m^{-3}$) with higher

values during the cold seasons (Oct–Jan) (average = 18±7 $\mu g\ m^{-3}$) and lower values during the warm seasons (Jun–Sep) (average = 13±3 $\mu g\ m^{-3}$) (Fig. 2a). Similarly, the HULIS concentration exhibited seasonal variations in the range of 13 to 294 RU (average = 108±77 RU) with higher values during the cold seasons (average = 152±76 RU) and lower values during the warm seasons (average = 46±20 RU) (Fig. 2b). After the HULIS content was normalized to the WSOC contents, the seasonal trend of the ratio of the HULIS content to the WSOC content was similar to that of the HULIS concentration, which

indicates that there was pronounced decrease in the average fraction of HULIS in WSOC from the cold to the warm seasons (Fig. 2c).

$\delta^{13}C_{WSOC}$, in the range of -21.0 to -27.5 ‰ (average = -24.0±1.5 ‰), exhibited no such seasonal variation trend (Fig. 2d). The levoglucosan concentration was in the range of 0.5 to 2.2 $\mu g\ m^{-3}$ (average = 1.1±0.5 $\mu g\ m^{-3}$) with relatively higher values

during the cold seasons and the lower values during Aug–Sep (Fig. 2e). The temporal variations in major ion species concentrations did not exhibit such seasonal behaviors throughout the year (Fig. 2f–2k). The temporal variation in $Ca^{2+}$ concentration (average = 0.8±0.2 $mg\ L^{-1}$) exhibited a relatively constant level throughout the year (Fig. 2f). The $SO_4^{2-}$ concentration (average = 12.0±10.3 $mg\ L^{-1}$) exhibited the highest value during the period of spring to summer and was slightly decreased during the cold seasons (Fig. 2g). The concentration of $NO_X$ (average = 11.7±8.7 $mg\ L^{-1}$) was high during

the warm periods (Jun–Oct) (Fig. 2h). The concentration of non-crustal K (average = 0.4±0.2 $mg\ L^{-1}$) also exhibited a relatively constant level throughout the year (Fig. 2i). The non-crustal V (average = 0.01±0.01 $mg\ L^{-1}$) showed no such seasonal variation (Fig. 2j). The sea-spray concentration ranged from 4.0 to 33.6 $mg\ L^{-1}$ (average = 17.3±6.4 $mg\ L^{-1}$) and showed the highest concentration in July (Fig. 2k). Both temperature and UV radiation rate gradually changed during the different seasons (Fig. 2l).


### 4 Discussion

In order to characterize the sources of WSOC and HULIS in Seoul, which may cause the seasonal variabilities, we analyzed various tracers including $\delta^{13}C_{WSOC}$, major ions, and molecular marker as source indicators (Kirillova et al., 2014b; Fu et al.,





2015; Kelly et al., 2005; Gabriel et al., 2002; Yan and Kim, 2015). The average $\delta^{13}C_{WSOC}$ (-24.0±1.5 ‰) suggests that the

burning activity of terrestrial C3 plant-origin materials could be a major source of WSOC in this region (Fu et al., 2015; Kelly et al., 2005). The average $\delta^{13}C_{WSOC}$ was in good agreement with that of HULIS (-25.4±1.6 ‰) extracted from precipitation in Seoul, Korea (Yan and Kim, 2017).

The good correlation ($r^2 = 0.6$; $p > 0.05$) between the WSOC and WSON concentrations suggests that these two variables are

highly associated with a common organic source (Yan and Kim, 2015) (Fig. 3a). The WSOC and HULIS concentrations also exhibited a good correlation ($r^2 = 0.5$; $p < 0.05$) (Fig. 3b). A good linear correlation ($r^2 = 0.5$; $p < 0.05$) was also observed between the HULIS and levoglucosan concentrations (Fig. 3c). This indicates that biomass burning was the major source of HULIS in Seoul, as levoglucosan is commonly used for the tracing of biomass burning (Kuang et al., 2015; Fu et al., 2015). This is consistent with the previous study, which has demonstrated that the BrC in the precipitation was primarily derived

from biomass burning and terrestrial biogenic emissions (>70 %), with minor contributions from fossil-fuel combustion, based on the $^{14}C$ values in the HULIS (Yan and Kim, 2017). However, we observed significant decreases (35±2 %) in levoglucosan concentrations by the UV irradiation for 12 to 24 h in the laboratory experiments (Fig. S5). This is consistent with the previous studies showing that levoglucosan could be oxidized by the hydroxyl radical in the atmosphere (Hennigan et al., 2010; Hoffmann et al., 2010). Thus, we conclude that the major source of HULIS was from biomass burning, but the

summer decreases in WSOC and HULIS concentrations could not be evaluated by using these tracers.

In order to indirectly evaluate the biomass burning during the study period, we compiled the fire maps in combination with air mass back trajectories (Fig. S1). An evidence for the biomass burning effect is provided by the fire maps, where higher occurrences of fire spots correspond to agricultural burning practices in the East Asia continents (Fig. S1). The burning

practices mainly occurred in the spring and summer in the East Asia continents. No significant influence of open burning activity was observed in the winter (Fig. S1). These results suggest that the distinctive biomass burning was not linked to the seasonal variations in WSOC and HULIS concentrations in this study region. On the other hand, non-crustal K, which is an indicator of biomass or fossil fuel burning (Baduel et al., 2010; Gabriel et al., 2002), did not exhibit a seasonal trend. The correlation between the HULIS and non-crustal K was also insignificant ($r^2 = 0.3$; $p < 0.05$) (Fig. 3d). The correlation

between HULIS and non-crustal V was also insignificant ($r^2 = 0.04$; $p < 0.05$) (Fig. 3e). The K/V ratio is often used to trace V-purified fossil fuels (Yan et al., 2012). The average K/V ratio was in the range of 17–673 (average = 124±154), which indicates that the V-purified fossil fuel contents in our samples were insignificant (Yan et al., 2012). Notably, no significant correlations were observed between the HULIS and $Ca^{2+}$ ($r^2 = 0.1$; $p < 0.05$), $NO_X$ ($r^2 = 0.1$; $p > 0.05$), $SO_4^{2-}$ ($r^2 = 0.02$; $p < 0.05$), and sea-spray component ($r^2 = 0.02$; $p > 0.05$) (Fig. 3f–3h), which indicates that the seasonal changes in HULIS

concentration were insignificantly influenced by crustal minerals, sea salts, and fossil fuels. Thus, we conclude the summer decreases in WSOC and HULIS concentrations were not associated with the changes in source inputs.





A significant negative correlation ($r^2 = 0.5$; $p < 0.05$) was observed between HULIS and the UV radiation (Fig. 3i). The HULIS was greatly reduced during the warm seasons (Jun–Sep) when the solar UV radiation and the temperature reached the annual maxima (Fig. 2b, 2l). It has been widely accepted that photochemical degradation is an important process for the efficient removal of chromophoric dissolved organic matter (CDOM) that absorbs and fluoresces at certain wavelengths of light under solar radiation in aquatic ecosystems, which alters absorption in the UV region, spectral shape, and CDOM composition (Helms et al., 2013; Mladenov et al., 2009). In addition, previous studies have demonstrated the significant photochemical degradation of CDOM in rainwater and further analysed the impact of the photo-resistivity of the BrC in atmospheric aerosols (Kieber et al., 2006; Kieber et al., 2007; Yan and Kim, 2017; Dasari et al., 2019). The WSOC and HULIS concentrations exhibited similar seasonal changes, while the seasonal changes in ratio of the HULIS concentration to the WSOC concentration are attributed to the larger changes in HULIS concentration during the warm seasons. These results are in agreement with the following laboratory experimental results.

In order to quantify the UV-degradable HULIS, we performed short-term (12 h) and long-term (42 d) UV radiation experiments. In the short-term test, aerosol samples ($N=48$) were exposed to the stimulated UV radiation for 12 h. More rapid degradations occurred in the winter samples with high HULIS concentrations, while no significant changes occurred in the summer samples with low HULIS concentrations (Fig. 4). The HULIS concentration was greatly reduced by 13% after the 12 h of UV irradiation. The WSOC concentrations of the cold season samples ($N=35$) exhibited followed corresponding decreases with time. However, negligible changes (<2 %) were observed in the summer samples ($N=13$). In the long-term test, two representative filter samples (summer and winter) were exposed to the UV radiation for 1 to 42 days. After the UV irradiation, in the winter sample, the HULIS and WSOC concentrations were decreased by approximately 52% and 25%, respectively (Fig. S6 and S7). However, no significant changes in HULIS and carbon contents occurred in the summer sample (Fig. S6 and S7). The UV radiation experiments showed that the fluorescence properties changed as the source HULIS was degraded by the UV radiation. This suggests that the summer sample might consist largely of a photo-refractory WSOC pool owing to the high UV radiation, while the winter sample consisted largely of a photo-labile WSOC.

The trend of light absorption of HULIS exhibited a consistent change with that of fluorescence properties during the UV radiation experiments (Fig. 5). The absorption coefficient of the summer sample was not changed after the irradiation, while that of the winter sample exhibited an approximately twofold change (Fig. 5). The degradation of HULIS was considerably more effective at shorter wavelengths ($S_{275–295}$), while no measurable change was observed at longer wavelengths ($S_{350–400}$), as expected from the fact that the absorption losses are higher in shorter wavelengths than the longer wavelengths (del Vacchio and Blough, 2002). In order to look at the changes in light absorption property, the spectral slope ratios ($S_R$) of shorter wavelength ($S_{275–295}$) relative to longer wavelength ($S_{350–400}$) were compared. In general, the $S_R$ value increases with increasing irradiation (Helms et al., 2008). If the input of fresh HULIS is constant throughout the seasons, $S_R$ should be constant. However, $S_R$ values of non-irradiated samples were 1.1 in the winter and 1.4 in the summer (Fig. 5). The $S_R$ value





of the non-irradiated summer sample was similar to that of the irradiated winter sample (1.4) (Fig. 5). Thus, the $S_R$ values suggest that the lower summer HULIS concentration was associated to the UV degradation rather than to the reduced source inputs.


## 5    Conclusions

The seasonal variations in optical and chemical properties of the HULIS and WSOC were monitored in the different seasons in the urban region. The air mass back trajectories combined with the fire maps and chemical analyses demonstrated that biomass burning was a major source contributing to HULIS and WSOC in all seasons in Seoul, Korea. Our results suggest
that photo-induced degradation plays a significant role in BrC quantity and quality in the atmosphere. In addition, the important role of photochemical degradation as a removal mechanism was confirmed by the laboratory experiments. This may provide important pieces regarding the hidden life cycle of light-absorbing organic aerosols in the atmosphere.

The light-absorbing organic aerosols are considerably associated with climate sensitivity owing to their significant roles in
the radiative forcing and global climate balance. Thus, the photochemical process needs to be considered in the modelling of climate forcing and biogeochemical cycles of the Earth's surface. Furthermore, the deposition fluxes of organic aerosols with different optical characteristics can have significant impacts as potential sources of organic carbon on surface waters, which can lead to major effects on the global carbon cycle.

*Author contribution.* H.Han and G.Kim involved in planning the research and designing the experiment. H.Han collected the data and performed the analyses and experiment. H.Han and G.Kim involved in analyzing the results and writing the manuscript. K.-H.Shin and D.-H.Lee contributed to the sample analysis of molecular compounds and writing the manuscript. All authors contributed to the final version of the manuscript.

*Competing interests.* The authors declare no competing financial interests.

*Acknowledgements.* This work was supported by the National Research Foundation (NRF) of Korea (NRF-2018 R1A2B3001147) funded by the Korean government. We are grateful to Jeonghyun Kim and Hojong Seo for their support and assistance. We also gratefully acknowledge the NOAA Air Resources Laboratory for use of the HYSPLIT model.

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






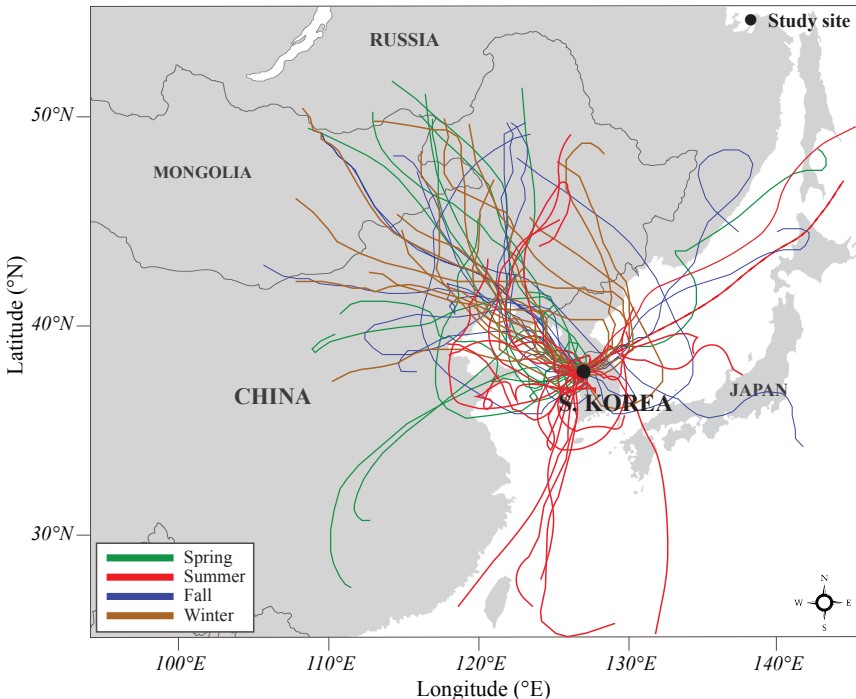

**Figure 1: Map of the geographical region around the study site and air mass transport pathways in the different seasons, spring (green), summer (red), fall (blue), and winter (brown). The ten-day air mass back trajectory was drawn by using the HYSPLIT model from March 2015 to January 2016 in Seoul.**







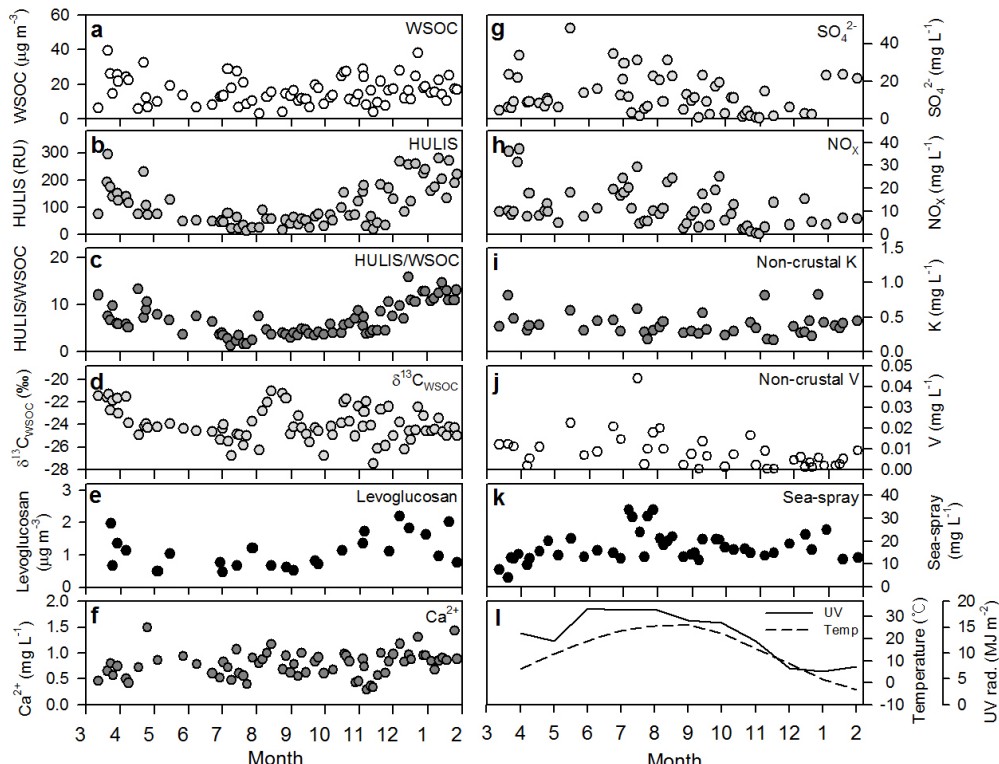

**Figure 2: Temporal variations in (a) WSOC concentration, (b) fluorescence intensity of HULIS, (c) ratio of HULIS to WSOC, (d)**
$\delta^{13}C_{WSOC}$ **values, (e) levoglucosan, (f) $Ca^{2+}$, (g) $SO_4^{2-}$, (h) $NO_X$, (i) non-crustal K, (j) non-crustal V, (K) sea-spray concentrations, (l)**
**UV radiation rate, and temperature from March 2015 to January 2016 in Seoul, Korea.**





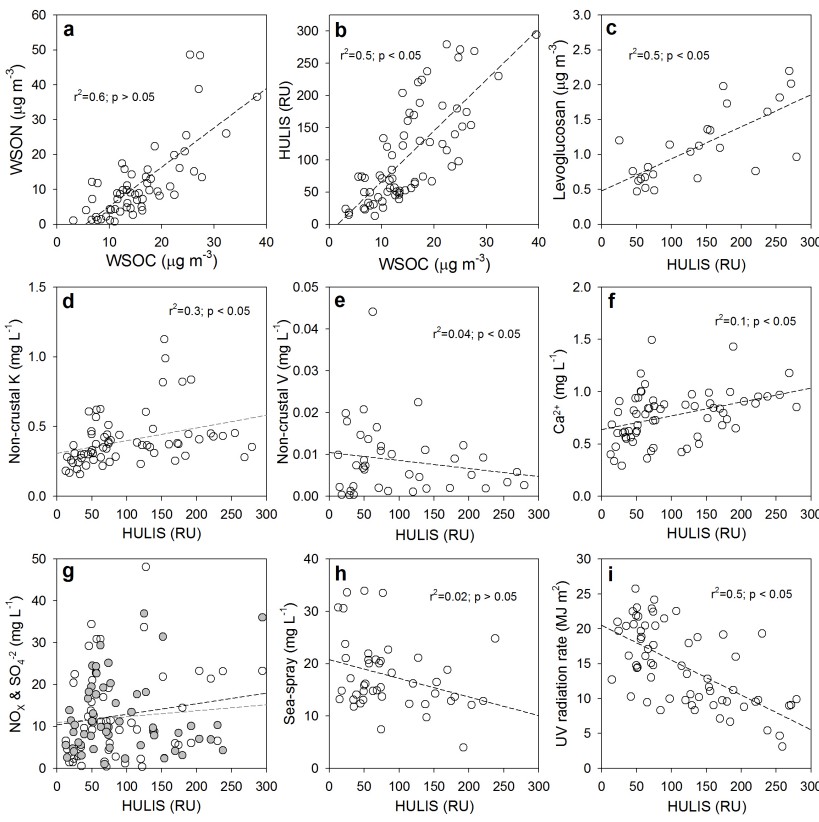

**Figure 3: Correlations between the concentrations of (a) WSOC and WSON, (b) WSOC and HULIS, (c) HULIS and levoglucosan, (d) HULIS and non-crustal K, (e) HULIS and non-crustal V, (f) HULIS and Ca$^{2+}$, (g) HULIS and NO$_X$ (closed circle; r$^2$=0.1, p>0.05) and SO$_4^{2-}$ (open circle; r$^2$=0.02, p<0.05), (h) HULIS and sea-spray, and (i) HULIS and UV radiation rate. The dashed lines represent the regression lines.**






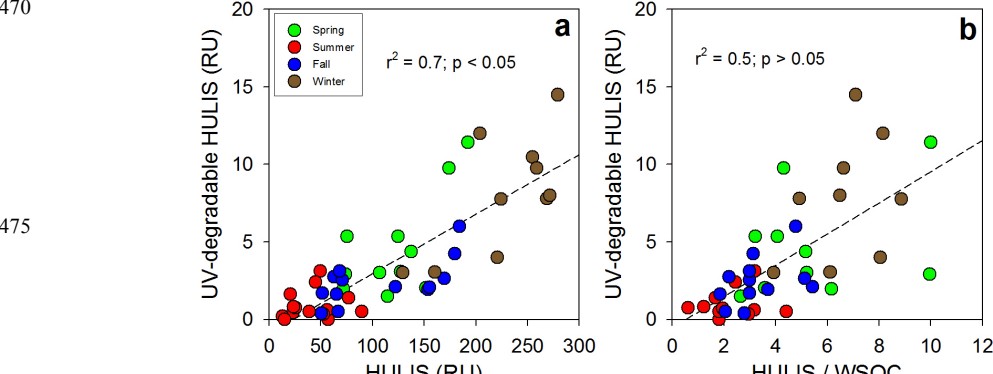


**Figure 4:** Correlations between (a) HULIS and UV-degradable HULIS and the (b) ratio of HULIS to WSOC and UV-degradable

HULIS over different seasons, spring (green), summer (red), fall (blue), and winter (brown). The dashed lines represent the regression lines.









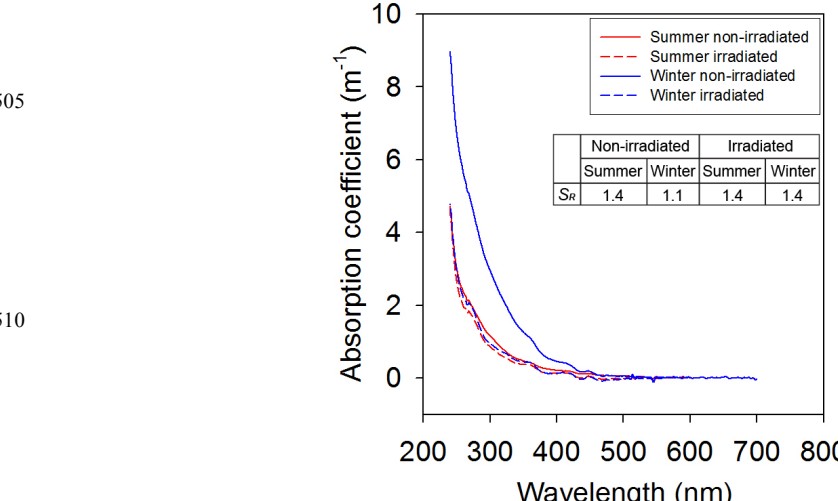

**Figure 5: The absorption coefficients of the aerosol samples collected in the winter (30.Dec) (blue) and summer (14.Aug) (red) in 2015. The solid lines represent the initial values for the non-irradiated samples, and the dashed lines represent the final values for the irradiated samples.**
