# Peer review of "Significant seasonal changes in optical properties of brown carbon in the mid-latitude atmosphere"

_Atmospheric Chemistry and Physics, 2019_

## Referee Comment (RC1) · Anonymous Referee #1 · 15 Dec 2019

General comments:

This paper presents the results of stationary measurements of biomass burning aerosol in South Korea over the course of multiple seasons. The authors identify biomass burning as a major contributor to light-absorbing components of the aerosol, and assert that UV-radiation can explain the lower absorptivity of summer vs winter samples, with irradiation of winter samples resulting in properties similar to observed summer samples. The paper is generally well-written and clear, and the figures are appropriate for the claims being made. In some cases, the language used is too conclusive, and some places need clarifying, as requested below.

Specific comments:

[Figure]

Abstract Line 10: seasonal changes in sources and sinks of HULIS and WSOC (what was directly measured rather than approximated). The next sentence then relates the HULIS to total BrC.

Abstract Line 17: this was support by laboratory UV radiation experiments (confirm is too strong of language)

Lines 23-24: Black carbon aerosol should not be included in the light absorbing organic aerosol category, as black carbon is primarily soot (inorganic carbon)

Lines 92-93: Since the K fraction and WSON are determined using parameterizations instead of direct measurements, "estimated" is a more accurate term than "calculated." For the K/Al_crust parameterization, please provide the value used in addition to the reference from where it came.

Line 95: The sea spray fraction is calculated assuming a certain ratio of Na to Ca, (1.4468:1) - how does this ratio compare to the measured Cl/Na ratio? Since the Cl and Na is assumed to come entirely from seawater, this assumption can be verified.

Line 98: Similarly, provide the crustal V/Al parameterization used, and replace "calculated" with "estimated."

Line 201: While Pearson's r-squared value of 0.5 does represent a decent correlation in atmospheric data, by definition this means that 50% of the variation in HULIS is attributable to levoglucosan, not a majority, so the phrase "was the major source" might be slightly misleading.

Line 228: Similarly, although the negative correlation with r-squared of 0.5 indicates a strong relationship between UV radiation and HULIS, it only explains half of the variance in HULIS, and the system is more complex than it sounds from the current language.

Line 242: the phrase " in the winter samples with high HULIS concentrations" leads one to conclude that there were winter samples with low HULIS as well as high HULIS.

If, as I assume, the authors simply want to reinforce the idea that winter had higher HULIS, then perhaps this would be clearer, "in the winter samples (characterized by higher average HULIS concentrations), while no significant changes occurred in the summer samples (characterized by low average HULIS concentrations)"

Lines 253-254: Please rephrase this sentence so that it is easier to understand. I don't quite understand what the authors are saying here.

Line 264: Given this interesting result (that irradiated winter samples look like summer samples" I wonder if there is more to be done with predicting summer properties from winter observations. It seems harder to back-calculate the properties of the aerosol emitted in summer (when still fresh), but I assume it would be similar to the winter samples. This interesting point deserves more attention, and I wonder if any previous studies have come to this conclusion, or a similar conclusion.

Line 271: The use of the word "confirmed" here to describe how the laboratory studies can explain the observed difference in fluorescence is too strong. The studies certainly support the hypothesis, and they go a long way toward suggesting the mechanism, but given that other processes in the atmosphere (oxidative, or possibly aqueous aging) could also result in the same bleaching, the language should be softened until there are many further studies, or until signatures of bleaching can be tied specifically to the mechanism studied here.

Technical corrections:

Line 21 "and THE global climate system"

Line 28 - remove plural on "light absorbing aerosols"

Lines 40-41 - remove space between number and percent symbol in all instances

Line 153: "originates"

Line 204: omit "the" before "BrC"

Line 244: there must be an extra word in this sentence "exhibited followed corresponding"

Line 248: remove plural on "carbon contents"

Line 251: omit "a" before "photo-labile"

Line 267: omit "the" before "different seasons"

---

## Referee Comment (RC2) · Anonymous Referee #2 · 17 Dec 2019

Han and coauthors describe off-line analysis of filter samples collected over four seasons in Seoul, Korea in order to investigate relationships between brown carbon (BrC) and water soluble organic carbon (WSOC) and how they change throughout the year. Potential sources are also discussed using complementary off-line techniques including high performance liquid chromatography (HPLC), excitation-emission matrix (EEM) analysis and isotope ratio mass spectrometry (IRMS). The seasonal behaviour of the relationship between humic like substances (HULIS) and WSOC is interesting, with a pronounced increase in the HULIS/WSOC ratio in the winter months. The authors propose that this pattern is due to more effective reduction of HULIS mas through photo-degradation during atmospheric transport in the summertime when photolyis rates and photochemical activity are higher. The absence of a seasonal pattern for potassium, in

particular adds weight to the argument that the HULIS/WSOC ratio seasonality cannot be explained only by changes in biomass burning source contributions throughout the year. Although levoglucosan exhibits higher wintertime values, this tracer can also be degraded through photochemical processing. Overall I find the manuscript to be well written and structured and only have some minor comments:

In the Introduction it is mentioned that "the origin of BrC is often attributed predominantly to HULIS". It would be helpful to discuss known sources of HULIS here.

Through EEM analysis, the extracted HULIS component agreed temporally with the extracted HULIS concentration as shown in S4. How do the other extracted EEM factors agree temporally with HULIS concentration? If the agreement is strong between all three factors, then it is likely that they are associated with the same source. If C2 is characterized by tryptophan then is vegetative debris a likely source for this factor? Are C1 and C3 interpreted to have the same source but differ only in their EEM characteristics? This section (3.1) could be clearer.

Either in the Introduction or Discussion it would be helpful to discuss the findings here in the context of previous on-line measurement studies that have observed similar photodegradation of brown carbon from biomass burning sources during atmospheric summertime transport, for example (Forrister et al., 2015; Selimovic et al., 2019; Healy et al., 2019).

Fix error in Formula 1

In Figure 3 many of these correlations are referred to as 'good' when 'moderate' would be more appropriate.

Line 154: "originates"

Define "RU"

Line 186: What is the likely source of the non-crustal V- oil combustion, shipping ? It would be useful to discuss.

Line 195: Define C3 plant-origin materials

Line 209-210: The last line of this paragraph is unclear and should be rephrased.

Line 244: Remove "followed"

References

Forrister, H., Liu, J., Scheuer, E., Dibb, J., Ziemba, L., Thornhill, K.L., Anderson, B., Diskin, G., Perring, A.E., Schwarz, J.P., Campuzano-Jost, P., Day, D.A., Palm, B.B., Jimenez, J.L., Nenes, A., Weber, R.J., 2015. Evolution of brown carbon in wildfire plumes. Geophysical Research Letters 42, 4623-4630.

Healy, R.M., Wang, J.M., Sofowote, U., Su, Y., Debosz, J., Noble, M., Munoz, A., Jeong, C.-H., Hilker, N., Evans, G.J., Doerksen, G., 2019. Black carbon in the Lower Fraser Valley, British Columbia: Impact of 2017 wildfires on local air quality and aerosol optical properties. Atmospheric Environment 217, 116976.

Selimovic, V., Yokelson, R.J., McMeeking, G.R., Coefield, S., 2019. In situ measurements of trace gases, PM, and aerosol optical properties during the 2017 NW US wildfire smoke event. Atmos. Chem. Phys. 19, 3905-3926.

---

## Author Comment (AC2) · 23 Dec 2019

Reviewer #2

Han and coauthors describe off-line analysis of filter samples collected over four seasons in Seoul, Korea in order to investigate relationships between brown carbon (BrC) and water-soluble organic carbon (WSOC) and how they change throughout the year. Potential sources are also discussed using complementary off-line techniques including high performance liquid chromatography (HPLC), excitation-emission matrix (EEM) analysis and isotope ratio mass spectrometry (IRMS). The seasonal behaviour of the relationship between humic-like substances (HULIS) and WSOC is interesting, with a pronounced increase in the HULIS/WSOC ratio in the winter months. The authors

propose that this pattern is due to more effective reduction of HULIS mas through pho-todegradation during atmospheric transport in the summertime when photolyis rates and photochemical activity are higher. The absence of a seasonal pattern for potas-sium, in particular adds weight to the argument that the HULIS/WSOC ratio seasonality cannot be explained only by changes in biomass burning source contributions through-out the year. Although levoglucosan exhibits higher wintertime values, this tracer can also be degraded through photochemical processing. Overall I find the manuscript to be well written and structured and only have some minor comments:

=> Thank you for your review and valuable comments. All your comments were ad-dressed in the revised manuscript.

In the introduction it is mentioned that "the origin of BrC is often attributed predomi-nantly to HULIS". It would be helpful to discuss known sources of HULIS here.

=> We added more about the sources of HULIS in the introduction section in the revised version.

Through EEM analysis, the extracted HULIS component agreed temporally with the extracted HULIS concentration as shown in S4. How do the other extracted EEM fac-tors agree temporally with HULIS concentration? If the agreement is strong between all three factors, then it is likely that they are associated with the same source. If C2 is characterized by tryptophan then is vegetative debris a likely source for this fac-tor? Are C1 and C3 interpreted to have the same source but differ only in their EEM characteristics? This section (3.1) could be clearer.

=> Since the other EEM factors are highly influenced by one dominant fluorophore HULIS (C1) as shown in the EEM spectra (Fig. S7 in the text; Fig. 1 below), we cannot reliably characterize C2 component using PARAFAC model. Thus, we only use C1 component in this study. This is mentioned in the text (lines 154–159).

Either in the Introduction or Discussion it would be helpful to discuss the findings here

in the context of previous on-line measurement studies that have observed similar pho-
todegradation of brown carbon from biomass burning sources during atmospheric sum-
mer time transport, for example (Forrister et al., 2015; Selimovic et al., 2019; Healy et
al., 2019).

=> Yes, this is mentioned in the introduction section in the revised version.

Define "RU"

=> The detailed definition about "RU" is added in the revised manuscript.

Line 186: What is the likely source of the non-crustal V- oil combustion, shipping? It
would be useful to discuss

=> Since vanadium is mostly removed during the refining processes, the use of raw
materials such as crude oil and coal can be the source of non-crustal vanadium. This
is briefly mentioned in the revised manuscript.

Line 195: Define C3 plant-origin materials

=> We briefly mentioned about C3 plant-origin materials in the revised manuscript.

Line 209-210: The last line of this paragraph is unclear and should be rephrased.

=> Yes, we rephrased this more clearly in the revised version.

Line 244: Remove "followed"

=> removed in the revised version.
———————————————————

[Figure]

**Winter (Dec.30.15)**

**Fig. 1.** Fig. S7

---

## Author Response (AR1)

**Final author response**

➔ Thank you for your review and comments concerning our manuscript entitled "Significant seasonal changes in optical properties of brown carbon in the mid-latitude atmosphere." We sincerely appreciate all valuable comments and suggestion. We thank all reviewers for taking the time and effort necessary to review our manuscript. We have concerned all your comments carefully, and all comments were addressed in the revised manuscript.

**Reviewer # 1**

General comments:
This paper presents the results of stationary measurements of biomass burning aerosol in South Korea over the course of multiple seasons. The authors identify biomass burning as a major contributor to light-absorbing components of the aerosol, and assert that UV-radiation can explain the lower absorptivity of summer vs winter samples, with irradiation of winter samples resulting in properties similar to observed summer samples. The paper is generally well-written and clear, and the figures are appropriate for the claims being made. In some cases, the language used is too conclusive, and some places need clarifying, as requested below.

➔ Thank you for your valuable comment. All your comments were carefully taken into account in the revised version.

Specific comments:
Abstract line 10: seasonal changes in sources and sinks of HULIS and WSOC (what was directly measured rather than approximated). The next sentence then relates the HULIS to total BrC.

➔ We measured both HULIS (water-soluble BrC) and MeOH-soluble BrC for all samples (Fig. 2b and Fig. S4b). For the UV radiation experiments, we only measured HULIS as an index of BrC. Thus, we kept the wording unchanged.

Abstract line 17: this was supported by laboratory UV radiation experiments (confirm is too strong of language)

➔ We changed the word "confirmed" to "supported."

Lines 23–24: Black carbon aerosol should not be included in the light absorbing organic aerosol category, as black carbon is primarily soot (inorganic carbon)

➔ We clarified this sentence in the revised version (lines 24–31).

Lines 92–93: Since the K fraction and WSON are determined using parameterizations instead of direct measurements, "estimated" is a more accurate term than "calculated." For the K/Al_crust parameterization, please provide the value used in addition to the reference from where it came.

➔ We changed the word "calculated" to "estimated." We also provide the values used for the parameterization, in addition to the reference in the revised version (lines 102–106).

Line 95: The sea spray fraction is calculated assuming a certain ratio of Na to Cl, (1.4486:1) – how does this ratio compare to the measured Cl/Na ratio? Since the Cl and Na is assumed to come entirely from seawater, this assumption can be verified.

➔ Here, 1.4486 is the ratio of the concentration of all elements except Cl to the Na concentration in seawater (Maenhaut et al., 2007). The mass ratio of Cl/Na is 1.79 for seawater. The measured Cl/Na mass ratio ranged from 0.01 to 1.78. Since the maximum sea-salt concentration in our samples was about 0.1% of seawater, the highest value (1.78) verifies that our assumption is valid. Since this is commonly referred to and sea spray fraction is negligible in this study, we do not explain this in the text.

Line 98: Similarly, provide the crustal V/A; parameterization used, and replace "calculated" with "estimated."

➔ We provided the value and changed the word "calculated" to "estimated."

Line 201: While Pearson's r-squared value of 0.5 does represent a decent correlation in atmospheric data, by definition this means that 50% of the variation in HULIS is attributable to levoglucosan, not a majority, so the phrase "was the major source" might be slightly misleading.

➔ We changed the word "major" to "significant" (lines 214–216).

Line 228: Similarly, although the negative correlation with r-squared of 0.5 indicates a strong relationship between UV radiation and HULIS, it only explains half of the variance in HULIS, and the system is more complex than it sounds from the current language.
➔ Yes, we agree. Thus, we just state similar trends for UV and HULIS. Then, we state that "these results are in agreement with the following laboratory results" instead of linking UV and HULIS at this stage.

Line 242: the phrase "in the winter samples with high HULIS concentrations" leads one to conclude that there were winter samples with low HULIS as well as high HULIS. If, as I assume, the authors simply want to reinforce the idea that winter had higher HULIS, then perhaps this would be clearer, "in the winter samples (characterized by higher average HULIS concentrations), while no significant changes occurred in the summer samples (characterized by low average HULIS concentrations)"
➔ We clarified this paragraph in the revised version (lines 252–255).

Line 253–254: Please rephrase this sentence so that it is easier to understand. I don't quite understand what the authors are saying here.
➔ We argue that the trend of "light-absorbing property" is similar to that of "fluorescence property." We clarified this sentence in the revised version (lines 261–262).

Line 264: Given this interesting result (that irradiated winter samples look like summer samples" I wonder if there is more to be done with predicting summer properties from winter observations. It seems harder to back-calculate the properties of the aerosol emitted in summer (when still fresh), but I assume it would be similar to the winter samples. This interesting point deserves more attention, and I wonder if any previous studies have come to this conclusion, or a similar conclusion.
➔ Thank you for your insightful comment. Although we cannot back-calculate the summer properties based on unknown UV radiation intensity, exposure time, and other atmospheric conditions, we assume a similar initial property based on other tracers including non-crustal K, air mass back trajectories (Fig. S1), and light absorption property in this study.

Line 271: The use of word "confirmed" here to describe how the laboratory studies can explain the observed differences in fluorescence is too strong. The studies certainly support the hypothesis, and they go a long way toward suggesting the mechanism, but given that other processes in the atmosphere (oxidative, or possibly aqueous aging) could also result in the same bleaching, the language should be softened until there are many further studies, or until signatures of bleaching can be tied specifically to the mechanism studied here.
➔ We changed the word "confirmed" to "supported."

Technical corrections:
Line 21: "and THE global climate system"
Line 28: remove plural on "light absorbing aerosols"
Line 40–41: remove space between number and percent symbol in all instances
Line 153: "originates"
Line 204: omit "the" before "BrC"
Line 244: there must be an extra word in this sentence "exhibited followed corresponding"
Line 248: remove plural on "carbon contents"
Line 251: omit "a" before "photo-labile"
Line 267: omit "the" before "different seasons"
➔ We made all above technical corrections as suggested.

**Reviewer # 2**

Han and coauthors describe off-line analysis of filter samples collected over four sea- sons in Seoul, Korea in order to investigate relationships between brown carbon (BrC) and water-soluble organic carbon (WSOC) and how they change throughout the year. Potential sources are also discussed using complementary off-line techniques including high performance liquid chromatography (HPLC),

excitation-emission matrix (EEM) analysis and isotope ratio mass spectrometry (IRMS). The seasonal behaviour of the relationship between humic-like substances (HULIS) and WSOC is interesting, with a pronounced increase in the HULIS/WSOC ratio in the winter months. The authors propose that this pattern is due to more effective reduction of HULIS mas through photodegradation during atmospheric transport in the summertime when photolyis rates and photochemical activity are higher. The absence of a seasonal pattern for potassium, in particular adds weight to the argument that the HULIS/WSOC ratio seasonality cannot be explained only by changes in biomass burning source contributions throughout the year. Although levoglucosan exhibits higher wintertime values, this tracer can also be degraded through photochemical processing. Overall I find the manuscript to be well written and structured and only have some minor comments:

➔ Thank you for your review and valuable comments. All your comments were addressed in the revised manuscript.

In the introduction it is mentioned that "the origin of BrC is often attributed predominantly to HULIS". It would be helpful to discuss known sources of HULIS here.

➔ We added more about the sources of HULIS in the introduction section in the revised version (lines 35–42).

Through EEM analysis, the extracted HULIS component agreed temporally with the extracted HULIS concentration as shown in S4. How do the other extracted EEM factors agree temporally with HULIS concentration? If the agreement is strong between all three factors, then it is likely that they are associated with the same source. If C2 is characterized by tryptophan then is vegetative debris a likely source for this factor? Are C1 and C3 interpreted to have the same source but differ only in their EEM characteristics? This section (3.1) could be clearer.

➔ Since the other EEM spectra are highly influenced by one dominant fluorophore HULIS (C1) as shown in Figure S7 (right), we cannot reliably characterize C2 component using PARAFAC model. Thus, we only use C1 component in this study. This is mentioned in the text (lines 165–170).

[Figure]

Either in the Introduction or Discussion it would be helpful to discuss the findings here in the context of previous on-line measurement studies that have observed similar photodegradation of brown carbon from biomass burning sources during atmospheric summer time transport, for example (Forrister et al., 2015; Selimovic et al., 2019; Healy et al., 2019).

➔ Yes, this is mentioned in the discussion section in the revised version (lines 244–247).

Define "RU"

➔ The detailed definition about "RU" is added in the revised manuscript (lines 133–136).

Line 186: What is the likely source of the non-crustal V- oil combustion, shipping? It would be useful to discuss

➔ Since vanadium is mostly removed during the refining processes, the use of raw materials such as crude oil and coal can be the source of non-crustal vanadium. This is mentioned in the revised manuscript (lines 231–234).

Line 195: Define C3 plant-origin materials

➔ We mentioned about C3 plant-origin materials in the revised manuscript (lines 205–207).

Line 209-210: The last line of this paragraph is unclear and should be rephrased.

➔ Yes, we rephrased this more clearly in the revised version (lines 221–222).

Line 244: Remove "followed"

➔ removed in the revised version.

[revised manuscript text omitted]

Author

850